# The Klinefelter Syndrome and Testicular Sperm Retrieval Outcomes

**DOI:** 10.3390/genes14030647

**Published:** 2023-03-04

**Authors:** Rosália Sá, Luís Ferraz, Alberto Barros, Mário Sousa

**Affiliations:** 1Laboratory of Cell Biology, Department of Microscopy, ICBAS-School of Medicine and Biomedical Sciences, University of Porto, Rua Jorge Viterbo Ferreira, 228, 4050-313 Porto, Portugal; 2UMIB-Unit for Multidisciplinary Research in Biomedicine, 4050-313 Porto, Portugal; 3ITR-Laboratory for Integrative and Translational Research in Population Health, Rua das Taipas 135, 4050-600 Porto, Portugal; 4Faculty of Medicine, University of Porto, Alameda do Prof. Hernâni Monteiro, 4200-319 Porto, Portugal; 5Department of Urology, Hospital Eduardo Santos Silva, Centro Hospitalar de Vila Nova de Gaia/Espinho, Rua Conceição Fernandes, 4434-502 Vila Nova de Gaia, Portugal; 6Department of Genetics, Faculty of Medicine, University of Porto, Alameda do Prof. Hernâni Monteiro, 4200-319 Porto, Portugal; 7Centre for Reproductive Genetics A. Barros, Av. do Bessa, 240, 1º Dto. Frente, 4100-012 Porto, Portugal; 8Institute of Health Research and Innovation (IPATIMUP/i3S), University of Porto, Rua Alfredo Allen, 208, 4200-135 Porto, Portugal

**Keywords:** Klinefelter syndrome, epidemiology, etiology, genetic causes, metabolic syndrome, testis, spermatogenesis, testicular sperm retrieval, predictive factors, newborn

## Abstract

Klinefelter syndrome (KS), caused by the presence of an extra X chromosome, is the most prevalent chromosomal sexual anomaly, with an estimated incidence of 1:500/1000 per male live birth (karyotype 47,XXY). High stature, tiny testicles, small penis, gynecomastia, feminine body proportions and hair, visceral obesity, and testicular failure are all symptoms of KS. Endocrine (osteoporosis, obesity, diabetes), musculoskeletal, cardiovascular, autoimmune disorders, cancer, neurocognitive disabilities, and infertility are also outcomes of KS. Causal theories are discussed in addition to hormonal characteristics and testicular histology. The retrieval of spermatozoa from the testicles for subsequent use in assisted reproduction treatments is discussed in the final sections. Despite testicular atrophy, reproductive treatments allow excellent results, with rates of 40–60% of spermatozoa recovery, 60% of clinical pregnancy, and 50% of newborns. This is followed by a review on the predictive factors for successful sperm retrieval. The risks of passing on the genetic defect to children are also discussed. Although the risk is low (0.63%) when compared to the general population (0.5–1%), patients should be informed about embryo selection through pre-implantation genetic testing (avoids clinical termination of pregnancy). Finally, readers are directed to a number of reviews where they can enhance their understanding of comprehensive diagnosis, clinical care, and fertility preservation.

## 1. Definition and Clinical Aspects 

Klinefelter syndrome (KS) is a congenital trisomy of male sex chromosomes, characterized by the presence of one or more extra X chromosomes [1,2], phenotypically presenting with testicular failure and hypergonadotropic hypogonadism [3,4,5,6,7].

About 80–90% of KS cases present a non-mosaic karyotype (47,XXY), while the remaining 10–20% evidence a mosaic karyotype (47,XXY/46,XY), higher grade aneuploidies (48,XXXY or 48,XXYY) or a structurally abnormal X chromosome (e.g., 47,iXq,Y), with mosaic cases originating a less severe KS phenotype [3,4,8,9,10,11,12].

KS is the most common chromosomal sexual anomaly and the most common chromosomal anomaly in men [3]. In a Danish population study of more than 2 million live births, the prevalence (existing cases) of KS was estimated to be about 0.15% (150:100,000 males) [13], varying with region and ethnicity [14], and the incidence (new cases) of KS was estimated to be around 1:500–1:1000 (0.1%–0.2%) male live births [7,9,13].

In its typical clinical picture, KS is phenotypically characterized by high stature (long limbs), small and hard testicles, small penis, gynecomastia (in late puberty), absence of body, pubic and facial hair, visceral obesity (feminine distribution of adipose tissue), eunuchoid body proportions (wide hips), and signs of androgen deficiency and infertility (mainly azoospermia) [15,16,17,18]. Patients with KS are diagnosed throughout their lives. Findings from a national survey of 200 patients with KS in Denmark showed that 20% were diagnosed prenatally, 35% were diagnosed during childhood, due to excessive growth and/or behavioral problems, and the remaining 45% were diagnosed in adulthood, typically as part of infertility evaluation [19].

In KS, hypogonadism is caused by a primary testicular disease. Hypergonadotropic hypogonadism is derived from structural and functional dysfunction of testicular Leydig cells (LCs) and Sertoli cells (SCs), and the hypothalamic–pituitary–gonad (HPG) axis. LCs present luteinizing hormone receptor (LHR) alterations, leading to luteinizing hormone (LH) resistance. This causes a subsequent decrease in testosterone (T) synthesis and secretion. Failure to respond to LH induces a state of elevated serum LH levels, which are thought to induce a compensatory LC hyperplasia; however, due to the small size of the testes and LC incompetence, LC hyperplasia does not compensate for the loss of androgen production. Normally, T exerts negative feedback over the hypothalamus, eliciting a decrease in gonadotropin-releasing hormone (GnRH) release. As GnRH acts over the adenohypophysis/anterior pituitary to induce the release of follicular stimulating hormone (FSH) and LH, this release is diminished. Additionally, T also exerts negative feedback on LH release from the adenohypophysis. These negative feedbacks are not observed in KS, which indicates the presence of a disrupted HPG axis, thus being also responsible for the rise in serum LH levels. Regarding the SC, these usually present receptors for androgens and FSH. By binding to the SCs, they elicit the synthesis and release of several molecules that act on the germ cells to induce and stimulate spermatogenesis. In KS, SCs do not respond to FSH, inducing a compensatory state of high serum FSH levels. FSH serum levels also remain high due to the SC decrease in the synthesis and release of inhibin-B, which would act on the adenohypophysis, where, by a negative feedback mechanism, it would lead to a decrease in the synthesis and release of FSH [20,21,22,23].

Low androgen levels are responsible for the observed androgen insufficiency (hypogonadism) in KS, with clinical features also dependent on other effects caused by an extra X chromosome [3,24]. Signs and symptoms may not always exist, their type and intensity vary, and their appearance also depends on the patient’s age, with the phenotype worsening with advancing age due to the cumulative influence of comorbidities [4,25,26,27,28]. Consequently, patients with KS exhibit a broad spectrum of phenotypes, with varying degrees of androgen deficiency severity [4]. Clinical characteristics of KS range from severe signs of androgen insufficiency to normally virilized men with just mild physical abnormalities, which explains why they do not receive clinical attention and are frequently underdiagnosed until adulthood, when they seek medical advice for infertility [10,12]. As a result, several patients who resort to infertility treatments are diagnosed here for the first time as having a 47,XXY karyotype. These patients have elevated FSH and LH serum levels but normal T serum levels. They do not exhibit signs and symptoms of decreased libido or androgenization, nor do they evidence endocrine, vascular, neurocognitive, or psychiatric disorders [29,30,31].

Newborn characteristics (length, weight and head circumference) are within normal limits, albeit slightly lower [30,32,33,34,35]. Subtle dysmorphic signs (fingers, palate, eyes) and congenital anomalies are uncommon and not pathognomonic [30,32,35,36]. They often present some degree of hypotonia, passive temperament, and developmental delays [30,35,37]. More frequently, genital anomalies (micropenis, cryptorchidia, bifid scrotum and hypospadias) are found [25,30,32,34,35,36,38,39,40,41,42]. These genital malformations are supposedly due to the effects of the extra X chromosome or androgen deficiency [3,4]. Longer legs [28,43,44] and speech disabilities [45,46] were attributed to the genetic abnormality rather than hypogonadism [3,4,16,39]. Small, hard testes are seen in all cases, and are caused by progressive tubulohyalinization and interstitial testicular fibrosis [1,3,4,16,30,39]. Hypogonadism also causes sexual dysfunction (decreased libido and erectile dysfunction) [3,4,47].

KS has multisystemic consequences, with higher morbidity and mortality than the general population, especially if not diagnosed and treated with T as early as possible [48]. Endocrine effects include osteoporosis, metabolic syndrome, obesity, and type 2 diabetes. It is also associated with musculoskeletal, cardiovascular and visual diseases, autoimmune disorders, cancer, neurocognitive disorders (learning disabilities), and behavioral disorders (psychosocial disturbances) [4,5,6,14,24,30,49,50,51,52].

Several metabolic disturbances are related to KS hypogonadism. Hypogonadism causes loss of muscle synthesis (causing a decrease in muscle mass), loss of glucose turnover (causing hyperglycemia), loss of fat turnover (causing fat deposition, truncal/visceral), and insulin resistance. Increased insulin resistance also produces LCs, aggravating the negative impact on T production [14,30]. Decreased muscle mass translates into decreased muscle strength and coordination, with hypotonia observed in most children. A metabolic syndrome develops as a result of an increase in fat mass, which is associated with increased serum levels of leptin (adipocyte-produced hormone) and ghrelin (gastro-pancreatic hormone that regulates hunger), hyperglycemia, increased insulin resistance (associated with development of type 2 diabetes), and dyslipidemia [14,30].

This metabolic state promotes the development of cardiovascular diseases, with a higher prevalence of atherosclerosis, arterial hypertension, myocardial infarction, stroke, arrhythmia, and heart failure. As T acts as a vasodilator, tissues tend towards a vasoconstrictor state due to decreased T levels, aggravating cardiovascular diseases. Low T levels and the metabolic syndrome, on the other hand, cause changes in hemostasis (clotting factors), which favor the development of thrombophlebitis, pulmonary thrombosis, and lower limb ulcers. The main hypothesis proposes that insulin resistance, obesity, and the metabolic syndrome increase plasminogen activator inhibitor (PAI) levels, conducting to a decreased fibrinolytic state (blockage of thrombus formation) [14,52,53]. PAI glycoprotein is a physiological inhibitor of tissue-type and urokinase-type plasminogen activators. Because these enzymes convert inactive plasminogen into plasmin, which then degrades fibrin, increased levels of PAI inhibit fibrinolysis, favoring thrombolysis [54,55].

Endocrine disorders also lead to abnormalities in bone metabolism, resulting in osteoporosis and tooth defects. In addition to hypogonadism, decreased lean body mass (all tissues and organs except fat body mass) and increased adiposity have been associated with low bone mass. Low bone mass is also associated with low levels of vitamin D (which ensures calcium ion absorption, facilitating calcium deposition in bones) in KS cases. Patients also evidence increased bone resorption. Bone resorption is mediated by the hormone insulin-like peptide 3 (INSL3), released by LCs in response to LH stimulation. INSL3 stimulates bone anabolism by acting on osteoblasts. However, individuals with KS have low serum levels of INSL3 due to LC insufficiency, which stimulates the secretion of sclerostin, which will exert catabolic effects on osteoblasts, favoring osteoporosis [56].

Patients with endocrine disorders and the metabolic syndrome are more likely to develop sexual dysfunction (erectile dysfunction and premature ejaculation), which can result in libido loss, anxiety, and depression [7,14].

Patients with KS may have neurocognitive deficits, including a decrease in intelligence quotient without loss of intellectual capacity, a delay in the development of complex learning and experience skills, and verbal deficits (verbal memory, processing speed, and vocabulary), which include language receptive skills (comprehension, ability to decode the meaning of words using auditory discrimination and processing, which depends on auditory capabilities and semantic memory) and expressive language skills (expression using the vocal motor part, which includes speech vocalization, articulation, and verbal fluency), deficits in reading and writing (dyslexia leads to comprehension and communication difficulties), and deficits in cognitive control processes (organization, planning, judging, decision-making, focused and sustained attention over time, inhibition of irrelevant information and information processing). Patients may have reduced attention, working memory, and cognitive flexibility, as well as a reduced capacity to evaluate facial expressions and form interpersonal relationships. Intriguingly, brain structure research demonstrated a reduced brain volume due to gray and white matter loss [14,30,57].

In general, patients report that the disease has a negative impact on their quality of life since childhood (physical, psychosocial, emotional, and social), pointing to decreased vitality and general health, insomnia, lower-paying jobs, greater difficulty in living with a partner, increased medication needs, and infertility [49].

In KS patients, various psychiatric illnesses such as schizophrenia, bipolar disorder, autism, attention deficit hyperactivity disorder, depression, and anxiety are more commonly documented [30,58,59].

KS is also associated with an increased prevalence of autoimmune diseases (Addison’s disease, type 1 diabetes, multiple sclerosis, rheumatoid arthritis, disseminated lupus erythematosus, and Sjögren’s syndrome) and cancer (breast, mediastinal, and hematological). Although no gene or genetic mechanism has been identified, this increase is most likely due to additional X chromosomal material, [14,30].

## 2. Etiology

The extra X chromosome originates from nondisjunction errors of the sex chromosomes, being 50% of paternal origin and 50% of maternal origin. Paternal nondisjunction of sex chromosomes occurs during meiosis I, whereas maternal nondisjunction can occur during meiosis I or II, or during early post-zygotic mitotic divisions [3,10,60]. In KS patients, the extra X chromosome is inactivated through epigenetic modifications [16,61].

## 3. Phenotype Variability

The presence of an extra X chromosome, androgen deficiency, and mosaicism are thought to influence the KS phenotype. Several mechanisms, including non-inactivation of specific genes, overexpression of somatic genes, altered methylation status, and altered androgen receptor sensitivity, have been proposed to explain the variability (severity) of the KS phenotype [14,62]. The phenotype will be more severe if these factors are attained at a larger level [63]. Mosaic KS patients have fewer genetic abnormalities, which results in milder clinical symptoms and endocrine abnormalities [64]. In contrast, the phenotype progressively worsens with the presence of more than one extra X chromosome [3,39,50], as with language and speech disabilities [4].

Specifically, several factors are associated with the KS phenotype. KS frequency rises with maternal and paternal age [65]. Patients with KS evidence gene dosage effects (X-linked gene duplication escaping inactivation), such as the short stature Homeobox containing gene on chromosome X (SHOX) gene, which is related to tall stature observed in KS [56,66]. The *SHOX* gene escapes X inactivation, is located in the pseudoautosomal region of the X and Y chromosomes, and is involved in skeletal development (the growth and maturation of bones in the upper and lower limbs) [66,67]. The *SHOX* gene effect can be enhanced since there are more copy number variants in regions expressing genes that also escaped X inactivation in KS patients [4,48,62,65,68,69].

Other mechanisms have been proposed, including the involvement of CAG (cytosine-adenine-guanine) repeats in the androgen receptor (AR) gene, FSH polymorphisms, and imprinting defects. The AR gene is located on the X chromosome and contains a highly polymorphic trinucleotide repeat known as the CAG region (as repeat length increases, androgen receptor activity decreases), which has been linked to physiological androgen effects [70,71]. When the number of CAG repeats is low, sensitivity to T increases. KS patients have a higher number of CAG repeats, resulting in lower T sensitivity, which correlates with the development of gynecomastia, small testes, and higher height due to decreased androgenicity [4,24,28,62,72]. Some KS patients exhibit FSH polymorphisms that might affect serum FSH activity [73]. Imprinting defects in sperm [74,75] and testicular germ cells [76] have been shown to be related with infertility [74,75,77]. KS patients also evidence germ cells with abnormal and variable DNA methylation of imprinted regions [78].

## 4. Spermatogenesis

KS is the most frequent chromosomal abnormality in infertile individuals (3–4%) and represents 8–12% of patients with secretory azoospermia [79,80,81,82,83]. Most KS patients are infertile (>99%) and have secretory azoospermia (91%) [3,14], while the remaining 8% present rare spermatozoa in the ejaculate (severe oligozoospermia or cryptozoospermia), rendering natural conception extremely difficult [3,14,24,60,84,85,86,87,88,89]. Mosaic KS patients seem to be less severely affected, with about half of them being azoospermic [64,90]. Before testicular tubular atrophy, patients can progress from severe oligozoospermia to hypospermatogenesis, maturation arrest, and tubules with only SCs, suggesting progressive spermatogenesis deterioration, with KS patients being grouped in three subtypes, those having focal spermatogenesis, those presenting focal incomplete spermatogenesis, and those having no germ cells [91,92,93].

Because KS patients are not a homogeneous group, the peripheral karyotype may not predict testicular cell chromosomal constitution or the presence or absence of spermatogenesis [94,95]. In accordance, several studies in KS patients, both mosaic and non-mosaic, evidenced testicular mosaicism in cases of focal spermatogenesis, with the presence of diploid (46,XY) and triploid (47,XXY) premeiotic germ cells.

Spontaneous pregnancies with birth of chromosomal normal children have been observed in mosaic KS patients [96] and non-mosaic KS patients [84], which was thought to be possible because of the presence of testicular mosaicism, as only 46,XY germ cells were considered capable of completing meiosis and XXY germ cells were considered meiotically incompetent. Subsequently, cytogenetic analysis of ejaculated spermatozoa from mosaic KS cases [97,98] and non-mosaic KS cases [99] revealed the presence of haploid and diploid spermatozoa, suggesting the presence of testicular mosaicism in instances with overt spermatogenesis. Although sperm analysis does not allow inferring the origin of haploid spermatozoa, the presence of diploid spermatozoa indicates that 47,XXY testicular germ cells can initiate meiosis and differentiate into spermatozoa. Testicular meiosis investigations confirmed the occurrence of testicular mosaicism in focal spermatogenesis. Some authors observed that 47,XXY premeiotic germ cells were not capable of originating spermatocytes and consequently spermatozoa, but only XY spermatogonia could support spermatogenesis. The presence of two distinct germ cell lineages was based on the observation that only 46,XY pachytene spermatocytes can be found in both in mosaic [100] and non-mosaic [95,100] KS patients.

In chromosomal normal men, 46,XY germ cells originate a similar proportion of 23,X and 23,Y spermatozoa. When 47,XXY spermatogonia can produce spermatozoa, the XX pairing will originate 24,XY and 23X spermatozoa, and the XY pairing will originate 24,XX and 23,Y spermatozoa in the same proportion. However, scientists discovered that the rate of 23,X spermatozoa in ejaculated spermatozoa from non-mosaic KS patients was higher than the rate of 23,Y spermatozoa, indicating a preferential pairing of homologous sex chromosomes (X favoring). These findings provided evidence that the majority of spermatozoa did not originate from 46,XY spermatogonia, a hypothesis that was further supported by the presence of a high proportion of spermatozoa showing numerical sex chromosome abnormalities, indicating that a large number of 47,XXY germ cells are capable of completing the spermatogenic process [99]. The authors then evaluated testicular germ cells in 10 non-mosaic patients, with 8 cases evidencing only SCs and 2 presenting foci of conserved spermatogenesis. Meiosis analysis in the 2 cases revealed a 47,XXY chromosomal constitution in SCs, spermatogonia, and primary spermatocytes, excluding testicular mosaicism. This evidences that spermatogonia could enter mitosis, originating 47,XXY spermatocytes and degenerative spermatocytes without sex chromosomes, instead of originating 46,XY/46XX and 45,X/45Y spermatocytes. As secondary spermatocytes, spermatids, and testicular spermatozoa evidenced a haploid and diploid chromosomal constitution, the results suggested that they originated from 47,XXY spermatogonia. As the 23,X spermatozoa rate was higher than the 23,Y spermatozoa rate, the results also indicated a preferential pairing of homologous sex chromosomes. In accordance, the 24,XY rates were higher than the 24,XX rates of secondary spermatocytes, spermatids and testicular spermatozoa [101].

Regarding the mechanisms underlying the presence of testicular mosaicism in focal spermatogenesis, the authors elaborated several hypotheses.

In non-mosaic KS, the authors suggested that in a subset of spermatogonia the extra X-chromosome was eliminated, allowing the development of 46,XY primary spermatocytes while the SC maintained a 47,XXY chromosome constitution [83,100,102,103]. Other authors observed that the majority of pachytene primary spermatocytes presented normal homologous recombination frequencies in non-mosaic KS, and that a slight majority (53%) of primary spermatocytes were 46,XY (testicular mosaicism). They thus hypothesized that, as altered homologous recombination increases the likelihood of aneuploidy, a synapsis defect might be associated with the higher rate of aneuploidy observed in KS. Although testicular sperm could be recovered and elicited the birth of healthy children, it was not possible to ascertain which type of spermatocytes produced the microinjected spermatozoon [104]. It was also hypothesized that the presence of normal 46,XY germ cell lines might be due to the repair of mitotic errors that occurred during primordial germ cells’ multiplication [105]. According to these observations, studies indicated that sperm found in the testes of men with KS have only a minor increased frequency of sex chromosomal polysomies (6–7%), which could also explain why most children born from fathers with KS have a normal karyotype [95,105]. The fact that most children born from KS fathers have a normal karyotype can be explained by the predominance of haploid spermatozoa in the ejaculate and testes; thus, the risk of transmitting numerical sex chromosome abnormalities appears to be relatively low and likely comparable to rates found in other severe male factor infertility patient groups [31,95,105].

Despite the presence of severe oligozoospermia or cryptozoospermia, spontaneous pregnancies and deliveries have been reported in mosaic KS patients [96]. Pregnancies were also observed in intracytoplasmic sperm injection (ICSI) treatment cycles using ejaculated sperm. These ICSI attempts yielded cases with absent pregnancy [106], abortion [107], and live births [108,109]. Spontaneous pregnancies and deliveries in non-mosaic KS patients have also been reported [84,110,111,112,113]. The use of ejaculated sperm in ICSI treatment [3,114] revealed cases without fertilization [115], with abortion [116,117,118], and live births [92,113,119,120,121,122,123,124].

## 5. Hormonal and Histological Characteristics

The testis is a gland with two major functions: spermatogenesis and hormone synthesis. The integrity of the testicular cells and of the HPG axis is required for proper testicular function. FSH and T receptors are not expressed in germ cells; hence, hormone effects are mediated through SC. Under FSH and T stimulation, SCs produce and secrete metabolites, hormones, and stimulating factors that act on germ cells to promote germ cell proliferation, meiosis, differentiation, survival, and apoptosis. However, in KS patients, spermatogenesis and hormone production functions are disrupted in consequence of LC, SC, and HPG axis dysfunction, with alterations in the pituitary feedback inhibition threshold being identified [123,125].

At the hormonal level, patients show high serum levels of estradiol (E2), FSH, and LH, decreased or normal levels of T, high levels of sex hormone-binding globulin (SHBG), formerly known as androgen-binding protein, low levels of INSL3, an LC product, low levels of inhibin B, a SC product, and low levels of anti-Müllerian hormone (AMH), a SC product [24,33,125,126,127,128,129,130]. A dysfunction of the LC, SC, and HPG axis explains why the LCs and SCs fail to produce T, INSL3, inhibin B, and AMH in response to the increased FSH and LH levels.

INSL3 receptors can be found on both LCs and germ cells. The synthesis of INSL3 by LCs is constitutive and thus independent of the cyclic LH stimulation. INSL3 interacts with specific receptors on LCs to modulate steroidogenesis, and the synthesis of INSL3 is modulated by T and LH (stimulate) and E2 (inhibits). Serum levels of INSL3 are consequently reduced in KS patients [126,128,129,131,132,133]. INSL3 is mostly involved in testicular descent and bone metabolism, but it also stimulates stem spermatogonia differentiation in the testes. As a result of the decreasing INSL3 levels in KS, these two functions are compromised [134]. High FSH and LH circulating levels are caused by failed responsiveness of SCs to FSH and of LCs to LH. This is exacerbated by low serum levels of inhibin B and T. Low serum inhibin B levels result in a loss of negative feedback over pituitary FSH production, and low serum T levels result in a loss of negative feedback over pituitary LH production, resulting in a low suppression of pituitary FSH and LH secretion. Furthermore, low serum T levels result in the loss of negative feedback over the hypothalamus, and consequently, GnRH continues to stimulate the release of FSH and LH from the adenohypophysis. Estradiol is produced as a result of T conversion by aromatase. The rise in E2 in KS patients has been attributed to a higher conversion at other sites due to the low levels of T in the testes. Other factors, such as decreased E2 clearance and weaker affinity to SHBG, have been proposed to contribute to the increase in E2 levels. Despite the high levels of E2, which acts as a negative feedback on FSH and LH pituitary release, the pituitary’s responsiveness remains unaltered, indicating a disrupted HPG axis in KS [3,15,24,27,135,136].

AMH induces Müllerian duct regression in males, preventing the development of the uterus and fallopian tubes, whereas T stimulates Wolffian ducts’ differentiation into vas deferens, epididymis, and seminal vesicles. The synthesis of AMH in immature SCs is stimulated by FSH and E2. FSH also stimulates SCs’ aromatase conversion of T to E2. Immature SCs have E2 and FSH receptors but not androgen receptors, therefore AMH levels remain high because T inhibits AMH synthesis. SCs mature and express androgen receptors after puberty, and an increase in T levels cause a decrease in AMH synthesis, which coincides with the establishment of the blood–testicular barrier (BTB) and meiosis, both of which are dependent on androgen action and SC maturity for the synthesis of cell junction proteins [126,133]. The insensitivity of LCs to LH and SCs to FSH in KS patients favors a decrease in T synthesis by LCs and intratesticular E2 production through FSH- induced SC aromatase conversion of T to E2. As a consequence, AMH levels fall to extremely low levels, and SC dysfunction, as well as low T levels, favor BTB disruption [126,129,133,135,136].

SHBG is primarily synthesized in the liver and serves as a transporter of androgens and estrogens in blood (preferentially to albumin) as well as a regulator of hormone availability to target tissues [137]. In KS, SGBH is elevated as a compensatory mechanism, resulting in a further reduction in T levels. Both T and E2 bind to SGBH at the same location, but in KS, SGBH binds preferentially to T, leaving more E2 free. In cases of KS with normal SGBH, this normalization of SGBH levels can be explained by low T levels, or that, because both T and E2 impact SGBH synthesis, decreased T levels and increased E2 levels would lead to the normalization of SGBH levels. E2 levels in patients with KS are thus increased due to an increase in peripheral conversion of T to E2, reduced metabolic clearance rates, and a poorer affinity for SGBH. However, an increase in E2 levels does not induce a decrease in LH and FSH production. In cases of increased levels of LH and normal T levels, the most plausible explanation is an elevation in the threshold for feedback inhibition of pituitary secretion [138]. As with T levels, there are KS cases in which E2 and SHBG levels may be normal [24,138]. Prolactin (PRL) levels have also been reported to be elevated [139].

The inadequate testicular environment favors the loss of SCs and germ cells, resulting in a reduction in spermatozoa production [132]. Despite LC hyperplasia, LCs are dysfunctional, not responding to LH, which leads to a decrease in the synthesis and secretion of T and INSL3. SC dysfunction, on the other hand, leads to a decrease in the synthesis and secretion of inhibin B and AMH [24,138]. This results in spermatogonia survival, proliferation, and differentiation being compromised [131,132]. Additionally, it is supposed that T transport may be hampered by an altered vascular network within the testicular interstice. KS patients, in fact, have decreased artery diameter [140,141]. Simultaneously, BTB disturbance would favor germ cell apoptosis [142,143].

At the histological level, a progressive testicular insufficiency is observed due to replacement of the interstitium by fibrosis and hyalinization of the seminiferous tubules. These alterations are responsible for the development of small and hard testes, as well as azoospermia. In the testicular interstitium, LC hyperplasia is observed, whereas in the seminiferous tubules, a gradual loss of spermatogonia is observed due to SCs’ massive phagocytosis and degeneration. Tubular cell degeneration leads to the sequential replacement of the normal testicular architecture by tubular atrophy (collapse), sclerosis (deposition of collagen components), fibrosis (stiffening caused by connective tissue replacement), and cell hyalinization (transformation of intra and extracellular proteins into homogeneous, vitreous, and pink material, with cellular degeneration) over time [8,17,132,143].

Several mechanisms have been proposed for the increasing depletion of germ cells, including insufficient extra X chromosome inactivation, LC deficiency, disrupted apoptosis control, and imprinting aberrations [7,8,78,132,144,145,146]. Germ cell loss is observed since fetal life [147,148], and it is aggravated throughout puberty when the HPG axis starts to function [149,150]. Apoptosis is thought to be caused by a deregulation of homologous recombination, with aneuploidy inducing non-homologous recombination and subsequent activation of apoptosis-related genes [60,151]. It was also claimed that testicular degeneration is caused by the loss of germ cells loss along with abnormal SC and LC maturation (146).

## 6. Testicular Sperm Retrieval

Male infertile individuals with severe oligozoospermia or secretory azoospermia may be able to father children with the development of ICSI [152,153,154] and testicular sperm extraction (TESE) [155,156,157,158,159].

The existence of focal spermatogenesis in KS patients also has allowed for successful spermatozoa recovery (SSR) from the seminiferous tubules [160,161,162]. The spermatozoa retrieval rate (SRR) in adolescents (15–19 y) and young adults (20–24 y) was 52%, 40–66% in adults and 30% in cases of cryptorchidism [31,89,163,164]. The kind of spermatogenesis failure with the SRR was specified in some instances. Although 38 (76%) of 50 KS patients had tubular sclerosis and atrophy, 9 (18%) had complete or incomplete germ-cell aplasia (only SC), and 3 (6%) had complete or incomplete maturation arrest, authors were able to retrieve spermatozoa in 24 (48%) of these cases. Unfortunately, authors did not specify the testicular phenotype in relation to successful sperm retrieval [165]. Another report of 47 KS patients revealed 34 cases with SCs alone (71% SRR), 9 with LC hyperplasia alone (33% SRR), 1 with maturation arrest (without sperm retrieval), and 3 with focal spermatogenesis (67% SRR) [166]. In another study of 45 KS patients, 58% of total SRR was obtained, with 29 (64.4%) cases presenting tubular sclerosis and atrophy (59% SRR), 12 (26.7%) presenting complete germ cell aplasia–LC hyperplasia (50% SRR), and 4 (8.9%) presenting maturation arrest (75% SRR) [167]. Another report of 6 KS patients with 35% of total SSR revealed 1 case with hypospermatogenesis and 5 cases with only SCs [168]. In a study of 9 KS patients, 6 presented only SCs (64% SRR), 2 presented foci of spermatogenesis (100% SRR), and 1 evidenced only LC hyperplasia (without sperm retrieval) [169]. These observations suggested that even in KS patients with a negative histopathological finding, a focus of spermatogenesis might be found.

### 6.1. Predictive Factors of Testicular Sperm Retrieval

Several markers were described in non-mosaic KS cases to characterize their androgen insufficiency and spermatogenesis dysfunction in comparison to the healthy population [128,136,170,171,172,173,174].

There are currently no clinical or biological parameters that can predict an SSR with certainty in these KS patients. However, several patient characteristics were shown to be related to SSR cases as compared to unsuccessful sperm retrieval cases.

Several studies found a significant increase in SSR with lower age [29,167,168,175,176,177,178,179,180,181,182,183], lower time of infertility [29], lower age, lower FSH levels, and high T levels [182], higher testicular volume and higher T levels [184,185], higher T levels [180,182,186], presence of 46,XY spermatogonia [103], higher androgen-binding protein levels [103], and with lower LH levels and higher T levels [181]. Other studies, on the other hand, did not find a significant increase in the SSR with regard to age, testicular volume (decreased), FSH (increased), and T (normal) levels [187,188], as well as in testicular echogenicity and intratesticular blood flow resistance [94]; age, testicular volume, FSH, LH (increased), and T (normal) levels, as well as in the FSH:LH ratio or in the androgen sensitivity index (LH x T) [165]; age, testicular volume, FSH, and T (decreased) levels [189]; testicular volume, FSH, LH, and T (low-normal) levels [175,190,191]; testicular volume, FSH, LH, and T (normal) levels [167,191]; age, testicular volume, FSH, LH, T (decreased), PRL (normal), E2 (normal) and inhibin (decreased) levels [192]; testicular volume, FSH, LH, T levels (low-normal), and PRL (normal) levels [168]; testicular volume, FSH, and inhibin B (decreased) values [178]; age, testicular volume, FSH, LH, and T (normal) values [193]; FSH and LH levels or serum inhibin B (decreased) levels [178,194]; FSH and LH levels [180]; or age, time of infertility, hormone levels, number of fragments at TESE, time of search at TESE [31].

Other studies were conducted to identify potential spermatogenesis markers, but without comparing cases with SSR versus cases with unsuccessful sperm retrieval. These included the findings of decreased testicular volume [195], decreased testicular volume, increased FSH and LH levels, and decreased T levels [162,196,197,198]; decreased testicular volume, high levels of FSH and LH, with normal T values [105,199,200]; increased levels of FSH and LH, slightly increased levels of PRL, with normal values of T and E2 [201]; increased levels of FSH and LH, normal levels of PRL and decreased T levels [202,203,204]; decreased testicular volume, high levels of FSH and LH, with normal PRL and T levels [205]; and decreased testicular volume, increased FSH levels, and decreased inhibin B levels [95].

The goal of T replacement therapy in young boys (early-to-mid-puberty) with KS is to promote linear growth, increase muscle mass, preserve bone density, and allow for the development of secondary sexual characteristics [206]. Although T replacement therapy improves symptoms of androgen insufficiency caused by KS, it also inhibits spermatogenesis at the spermatogonia stage in adults. Furthermore, while T may aid in LH suppression, LC synthesis of the germ cell protector INSL3 will be decreased [3,33]. Adult T replacement therapy should thus be used solely in patients who are not interested in infertility treatments and have androgen deficiency. It should be discontinued for at least 4–6 months prior to infertility treatment if used [172,173].

### 6.2. Techniques of Testicular Sperm Retrieval

Many KS patients have sought infertility treatment as adults, with no other symptoms or signs besides infertility. These patients had decreased testicular volume, high FSH and LH mean serum levels, and normal or slightly lower T concentrations. This group of patients did not belong to the main group (65–85%) of KS patients reported as having low T levels [171].

For patients, the testicular SRR is critical. Previous reviews showed a mean SRR of 44% [9], with a range of 30–70% [10]. This observed variability in the SRR suggests a possible effect of the different number of patients studied, retrieval technique, and differences in patient characteristics.

According to studies, non-mosaic KS patients with low T levels should be treated with aromatase inhibitors first to decrease E2 levels and therefore enhance T intratesticular availability. This was suggested to potentially improve spermatogenesis in KS cases with foci of spermatogenesis [172]. When applied to men with low T levels or low T:E2 ratios, a higher retrieval rate (66%) has been reported using pre-treatment with aromatase inhibitors to equilibrate the T:E2 ratio [162,166,177]. However, these success rates could also be attributed to the simultaneous use of microsurgical testicular sperm extraction (mTESE). Because of the scarcity of research in KS patients and the absence of comparisons between conventional TESE (cTESE) and mTESE in patients treated with aromatase inhibitors, this kind of treatment is not routinely followed in cases with low T levels and high FSH and LH values. In fact, there have been cases of SSR after cTESE in patients with slightly low T levels who did not receive aromatase pre-treatment (29, 31). Nevertheless, aromatase treatment should be administered in those circumstances [177,182].

The mTESE [207] is a very promising TESE procedure, presenting high rates of SSR (47–69%) in KS patients [166,167,175,176,177,179,185,190,192,198,200,208,209], though none of these reports compared mTESE to cTESE. Although not in KS patients, both approaches were compared in the pioneering work of Schlegel [207,210], with authors obtaining an SRR of 63% by mTESE vs. an SRR of 41% by cTESE. There have been very few reports comparing cTESE to mTESE in KS patients. A study compared 28 cases using cTESE (50% SRR) to 10 cases with mTESE (10% SRR) [193]. Another report compared 23 cases using cTESE (0% SRR) to 20 cases with mTESE (33% SRR) [186]. A third report compared 43 cases using cTESE (51% SRR) to 40 cases with mTESE (33% SRR) [187].

These extreme rates were not anticipated. As a result, we conducted a thorough literature review that included 63 reports (Table 1). The overall SRR was 43% (777 SSR/1809 patients). In total, 37 reports used cTESE, 18 reports used mTESE, 5 used both mTESE and cTESE but did not provide results per technique, 2 used fine needle aspiration, and one report used testicular sperm aspiration. Neither FNA nor TESA are the indicated techniques for testicular sperm retrieval due to the presence of random focal spermatogenesis in KS patients.

In cTESE (37 reports), the SRR was 44% (228/516), with a range of 16–100%. There were 3 reports with ≥50 KS cases (SRR of 38, 40, and 48%) [29,31,165], with none presenting an SRR greater than 50%. Of the other 34 cases, there were 16 case reports with 100% SRR [20,23,88,104,157,162,197,198,201,202,203,204,205,211,212,215], one with 4 patients and 75% SRR [95], one with 18 cases and 28% SRR [194], one with 19 cases and 21% SRR [94], another with 5 patients and 20% SRR [218], one with 25 patients and 16% SRR [189], and 13 reports exhibited an SRR of 30–57% [103,105,114,115,118,160,161,163,168,178,184,213,214].

In mTESE (18 reports), the SRR was 43% (427/991), with an SRR range of 17–100%. There were 7 reports with ≥50 KS cases (SRR of 20, 28, 33, 43, 47, 57, and 66%) [167,177,180,181,182,185], of which 2 cases presented an SRR of more than 50% (57, 66%) [167,177]. The other 11 KS cases evidenced SRRs ranging from 17% [191] to 40–74% [166,169,179,192,208,217,219,220], with 2 cases displaying a 100% SRR, one with 2 patients [200] and the other with 9 patients [216].

The mTESE requires a surgical unit and specialized equipment, is aggressive (the testis is completely opened transversally), complex, time-consuming, and expensive. Many authors perform cTESE in a surgical unit, and the procedure is also time-consuming and expensive, as multiple testicular openings are performed if spermatozoa are not found [183]. A very efficient modified cTESE procedure, on the other hand, involves spermatic cord block (local anesthesia) in an outpatient setting (without a surgical unit), with a single 1 cm scrotum excision to reach the tunica vaginalis space. Thereafter, a 0.5 cm incision is made to expose the seminiferous tubules, followed by a biopsy of a small fragment (1–2 mm) that is immediately examined for spermatozoa. Wherever necessary, the testis is simply rotated exposing its different faces, eliminating the need for additional scrotal incisions. The procedure takes about 30–45 min and no complications were observed [29,31]. cTESE, like mTESE, requires the services of a highly skilled experienced urologist (Figure 1).

Recently, comparisons of both methods, cTESE and mTESE, were conducted. A meta-analysis involving 1248 KS patients found no statistically significant differences in SSR between the two methods, with a mean SRR of 44% (43% in cTESE, 45% in mTESE, and 41% in mixed cases) [188]. A more recent comprehensive review of the literature on KS patients also revealed a mean SRR of 42–57% using both testicular sperm retrieval procedures [221]. These differences highlight that SRR variability may be attributable to differences in patient characteristics as well as differences in the number of patients analyzed or the retrieval technique. Nevertheless, it would be relevant if the major American and European groups could conduct a prospective comparative study of mTESE vs. cTESE in a large number of KS cases.

### 6.3. Clinical and Newborn Outcomes

Several previous studies reported the use of cryopreserved testicular spermatozoa [95,115,120,168,169,182,194,200,208,215], but the small number of cases prevented comparisons. Other studies comparing fresh with cryopreserved testicular spermatozoa reported similar rates of clinical pregnancy (CP) and newborns (NBs) [115,168,193,209], whereas other studies have found lower CP and NB rates in cryopreserved testicular spermatozoa [29,31].

Some studies provided larger numbers of patients enabling comparisons between fresh and cryopreserved spermatozoa treatment cycles. There were 10 fresh and 16 cryopreserved testicular spermatozoa treatment cycles in a report of 38 KS patients with an SRR of 39% using cTESE and mTESE. In ICSI cases with cryopreserved testicular spermatozoa, the authors observed a significantly higher embryo cleavage rate (ECR) [193]. There were 20 fresh and 17 cryopreserved testicular spermatozoa treatment cycles in a report of 65 KS patients with an SRR of 39% using cTESE. In ICSI cycles using fresh testicular spermatozoa, the authors reported a significantly higher fertilization rate (FR), number of high-grade embryos, and CP rate [29]. There were 32 fresh and 12 cryopreserved testicular spermatozoa treatment cycles in a study of 83 KS patients with an SRR of 42% using cTESE and mTESE. The implantation rate (IR) was significantly higher in ICSI cases with cryopreserved testicular spermatozoa [187]. There were 25 fresh and 22 cryopreserved testicular spermatozoa treatment cycles in a recent report on 77 KS patients with a 40% SRR using cTESE. In ICSI cycles using fresh testicular spermatozoa, authors observed a higher FR, number of high-grade day-3 embryos, biochemical pregnancy rate, CP rate, IR and live birth delivery rate (LBDR) [31]. In conclusion, two studies indicated that ICSI cycles using cryopreserved testicular spermatozoa had higher ECR and IR, while two studies revealed that ICSI cycles using fresh testicular spermatozoa had higher FR, ECR, number of high-quality embryos, IR, CP rate, and LBDR.

The authors of a meta-analysis of cases with secretory azoospermia first concluded that there were no significant differences in embryological and clinical outcomes using ICSI either with cryopreserved testicular spermatozoa or with fresh testicular spermatozoa. However, the authors also confirmed that when different pathologies, such in KS cases, were individually examined, fresh testicular sperm produced better outcomes [222]. Thus, it is possible that the observations that using cryopreserved testicular spermatozoa yielded higher rates of embryo cleavage and implantation [187,193] depends on a bias caused by the fact that only two parameters were evidenced and the number of cases analyzed was relatively low, with this data suggested to be inconclusive.

In terms of NB, the current updated literature analysis (Table 1) reveals 9 reports with more than 10 NBs and 6 with more than 20 NBs, for a total 47% NB rate per embryo transfer cycle (ETC) (315 NB/669 ETC). The karyotype was normal in all of these cases.

There has been some concern about the possibility of an increased risk of chromosome abnormalities in the offspring of KS patients, after increased rates of both autosomal aneuploidies [16,99,105,223,224,225,226,227] and sex-chromosome aneuploidies were found in spermatozoa from KS men, albeit at a low rate [228], and in preimplantation embryos [114].

Up to the present, all children born after using testicular sperm from KS patients present a normal karyotype. This may be explained by previous observations that indicated that the majority of the 47,XXY germ cells are not meiotically competent, being thus unable to originate chromosomal abnormal sperm [83], and by other reports that observed a normal pattern of sex chromosome segregation in most of the testicular sperm retrieved from KS patients [105].

Until the present day, all published data have revealed that, with the exception of two cases, none of the children born from KS patients exhibited an abnormal chromosomal constitution [29,31,176,177,185,187,188,193,221,229]. The prevalence of chromosomal abnormalities in the general population has been found to be at a 0.5–1% rate [230,231]. The current research demonstrates that, as only two children (0.63%) were afflicted in 315 children, the risk of Klinefelter karyotype transmission is low. Thus, the present data reassures that KS men have no increased risk of transmitting their genetic problem to offspring. Nonetheless, patients with KS should be informed of the technical possibility of further genetic diagnosis procedures, even though the benefit of preimplantation genetic testing (PGT) or prenatal diagnosis (PND) is questionable in the light of these findings.

## 7. Final Considerations

Population studies are not entirely consistent when it comes to the frequency of KS diagnosis per age groups. In an English population research, Abramsky and Chapple [232] found that the diagnosis was made in 1% of cases aged < 1 years, 0% of cases aged 1–10 years, 7% of cases aged 11–20 years, and 19% of cases aged > 20 years. In a Danish population, Bojesen et al. [13] observed that the diagnosis was made in 6.5% of cases aged 1–10 years, 14.2% of cases aged 10–14 years, 24.9% of cases aged 15–19 years, and 54.4% of cases aged > 20 years.

Swerdlow et al. [233] found in a study of an English population that the diagnosis was made in 21.5% of cases between 0–14 years, 22.5% of cases between 15–24 years, and 56% beyond 25 years. In a Danish population, Aksglaede et al. [19] discovered that 20% were diagnosed prenatally, 35% were diagnosed during childhood due to excessive growth and/or behavioral problems, and the remaining 45% were diagnosed in adulthood, mainly as part of infertility evaluation.

In an Australian population study, Herlihy et al. [234] reported that the frequency and main causes of diagnosis by age were: newborn (0–1 years), infancy (3–5 years) and childhood (6–10 years): cryptorchidism (27–37%), delayed speech (>40%), learning difficulties (>75%); adolescence (11–19 years): decreased T levels (63–85%), decreased facial (60–80%) and pubic (30–60%) hair, gynecomastia (38–75%), small testes (>95%), and infertility (>99%); and adulthood (>20 years): decreased T levels (63–85%), decreased facial (60–80%) and pubic (30–60%) hair, metabolic syndrome (46%), type 2 diabetes (10–39%), osteopenia (~40%) and osteoporosis (10%), small testes (>95%), and infertility (>99%). In conclusion, the diagnosis is usually made up to the age of 10 years due to the presence of growth and behavior problems, and in adolescence mainly due to the absence of virilization, while the remaining cases elapsed without a diagnosis of KS, being able, however, to suffer from morbidities associated with KS, unless they went to infertility centers.

During the prepubertal period, reproductive counseling is usually not provided, and children with physical–cognitive deficits must be properly supported like any other child with the same problems and without a Klinefelter karyotype. In the absence of signs of masculinization (hair, penis, voice) and the presence of androgen deficit (small penis, gynecomastia, small testicles), from the age of puberty, the andrologist may consider introducing corrective treatment for the androgen deficit towards the end of puberty (16–18 years) depending on the clinical picture.

The adolescent should be informed that the vast majority of KS patients are infertile, with spontaneous pregnancies occurring quite infrequently [54,96]. If the adolescent is capable of erection and ejaculation prior to therapy, a semen analysis will reveal if he has spermatozoa in the ejaculate or azoospermia. The use of electrostimulation should be discussed in the absence of an erection. Prophylactic sperm cryopreservation should be offered in the presence of spermatozoa in the ejaculate, because KS has a gradual pattern of germ cell loss [132]. Based on past findings, the hypothesis of testicular sperm collection should be discussed in the context of azoospermia. In this regard, adolescents (15–19 y) and young adults (20–24 y) had an SSR of 52%, while adults had an SSR of 40–66%, and cryptorchidism had an SSR of 30% [31,89,163,164]. This indicates that TESE may be not necessary in adolescents with azoospermia. Additionally, in the presence of small testicles, TESE may negatively impact on the testicular milieu [235]. As a result, it was advised that it would be best to abandon the TESE approach for the time being. Some authors considered the possibility of cryopreserving testicular tissue or isolate spermatogonia stem cells in cases of absent testicular sperm after TESE and in prepubertal boys [143,236], but these approaches are still experimental and not recommended [235].

Some evidence indicates that fertility preservation should perhaps not be offered to KS patients under the age of 16 [237], and that androgen treatment should begin at the age of 18 [238], as most patients have absent or minimal androgen deficiency before that age [239]. Although androgen therapy should be started as soon as possible [3], clinicians should keep in mind that it impairs spermatogenesis [177]. When fertility preservation prior to androgen treatment is not possible, authors advised using aromatase inhibitors or human chorionic gonadotropin (HCG) to preserve spermatogenesis [166,217]. Additionally, once the desired outcomes have been achieved, androgen treatment can be temporarily halted while the patient is monitored continuously.

There are several techniques for retrieving epididymal and testicular sperm, including percutaneous epididymal sperm aspiration (PESA), microscopic epididymal sperm aspiration (MESA), fine needle testicular aspiration (TESA), testicular sperm extraction through open biopsy procedure (cTESE), and microsurgical seminiferous tubule identification with microsurgical testicular sperm extraction through wide opening of the testis (mTESE). PESA, MESA, and TESA can be used to treat cases of obstructive azoospermia, with TESA being the most commonly used technique due to the better clinical outcomes. It is necessary to employ cTESE or mTESE for secretory azoospermia (non-obstructive azoospermia), which occurs in KS. The advantages and disadvantages of cTESE and mTESE techniques have already been highlighted. In KS patients, testicular attainment causes secretory azoospermia rather than obstructive azoospermia, with cTESE or mTESE employed for fertility treatments. Patients with decreased serum T levels are additionally indicated for aromatase inhibitor treatment before testicular sperm extraction, and any testicular sperm collection should take place at least six months after terminating T medication.

All of these issues are delicate and require multidisciplinary specialized guidance, which is why some physicians prefer to discuss these options at the age of 18.

We believe that this study will serve as a clinical guide, allowing children, adolescents and adults with the Klinefelter karyotype to be diagnosed earlier, properly referred and treated, avoiding significant systemic complications, and preserving fertility in a timely way.

We hope that this review will encourage each country’s health system to publish population-based studies on KS in order to improve data on incidence and prevalence, as well as to uncover the reality of systemic complications, indicating the usefulness of better including KS in the differential diagnosis of several conditions, and thus for them to suggest karyotyping.

The outstanding clinical results obtained with testicular spermatozoa and the absence of babies born with the Klinefelter karyotype documented here confer the TESE technique a high level of confidence. Although the option for cTESE or mTESE depends on the experience of the clinical team and the resources available, and although large systematic reviews indicate proximity in clinical results, it is evident that more comparative research with a large number of cases is needed.

## Figures and Tables

**Figure 1 genes-14-00647-f001:**
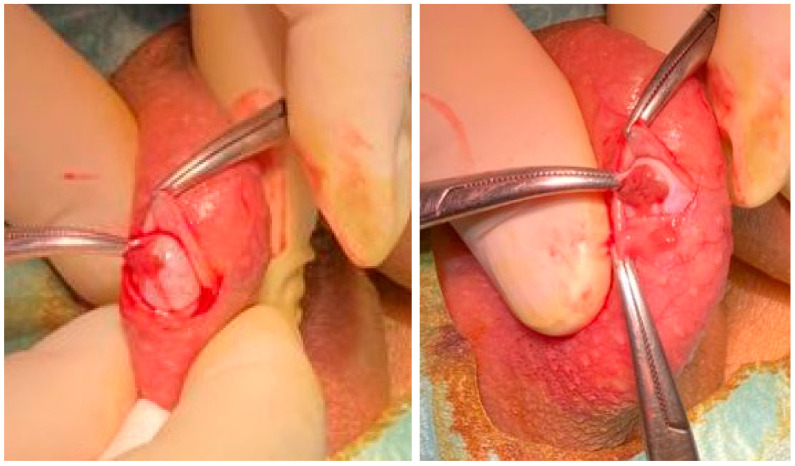
Outpatient conventional TESE employing local anesthesia using the three-finger technique and spermatic cord block. **Left**. Exposure of the tunica albuginea after entering the tunica vaginalis space. **Right**. Exposure of the seminiferous tubules.

**Table 1 genes-14-00647-t001:** Literature review on sperm retrieval and newborn outcomes in non-mosaic Klinefelter syndrome patients.

References	Pacients (n)	SSR (n)	SRR (%)	ETC (n)	CP (n)	NB (n)	Technique
[211]	3	3	100	4	0	0	cTESE
[157]	1	1	100	1	0	0	cTESE
[160]	9	4	44	3	0	0	cTESE
[161]	15	8	53	7	2	2	cTESE
[162]	2	2	100	3	3	3	cTESE
[199]	7	4	57	4	4	1	FNA
[101]	10	2	20			DTB	FNA
[201]	1	1	100	1	1	2	cTESE
[202]	1	1	100	2	1	1	cTESE
[105]	20	8	40	8	4	7	cTESE
[120]	1	1	100	1	1	2	cTESE
[212]	1	1	100	4	3	3	cTESE
[115]	12	5	42	10	5	6	cTESE
[203]	1	1	100	1	1	2	cTESE
[205]	1	1	100	1	1	1	cTESE
[204]	2	2	100	2	2	2	cTESE
[94]	19	4	21	4	4	3	cTESE
[95]	4	3	75	4	1	1	cTESE
[184]	20	9	45			DTB	cTESE
[196]	1	1	100	2	1	1	TESA
[103,213]	24	12	50	4	4	5	cTESE
[114]	20	10	50	26	8	5	cTESE
[118]	11	6	55	15	2	1	cTESE
[194]	18	5	28	5	2	2	cTESE
[214]	14	8	57	8	6	9	cTESE
[123]	1	1	100	3	1	1	cTESE
[189]	25	4	16	2	2	1	cTESE
[165]	50	24	48			DTB	cTESE
[104]	4	4	100	3	3	6	cTESE
[175]	51	26	51	26	12	12	mixed
[208]	10	6	60	8	6	3	mTESE
[166]	42	29	69	29	18	22	mTESE
[167]	74	42	57			DTB	mTESE
[197]	1	1	100	2	1	1	cTESE
[192]	26	13	50	13	4	2	mTESE
[168]	17	6	35	9	7	8	cTESE
[200]	2	2	100	3	2	3	mTESE
[215]	2	2	100	2	2	1	cTESE
[216]	9	9	100			DTB	mTESE
[178]	27	8	30	8	4	7	cTESE
[177]	68	45	66	62	33	28	mTESE
[179]	39	22	56	18	7	2	mTESE
[176]	106	50	47	49	26	29	mTESE
[198]	1	1	100	1	1	1	cTESE
[93]	69	33	48			DTB	mixed
[193]	38	15	39	23	15	16	mixed
	28	14	50				(cTESE)
	10	1	10				(mTSE)
[217]	10	7	70			DTB	mTESE
[218]	5	1	20			DTB	cTESE
[191]	18	3	17	3	1	1	mTESE
[29]	65	25	38	36	16	17	cTESE
[180]	134	38	28	18	4	5	mTESE
[169]	9	6	67	6	1	1	mTESE
[163]	41	23	56			DTB	cTESE
	25	13	52				(15–22 y)
	16	10	63	10	4	3	(≥23 y)
[181]	135	46	33			DTB	mTESE
	10	1	10				(13–14 y)
	50	19	38				(13–19 y)
	85	26	31				(20–61 y)
[186]	43	6	14	6	3	5	mixed
	23	0	0				(cTESE)
	20	6	30				(mTSE)
[219]	10	5	50			DTB	mTESE
[187]	83	35	42	41	22	25	mixed
	43	22	51				(cTESE)
	40	13	33				(mTSE)
[220]	5	2	40	2	1	1	mTESE
[182]	110	22	20	31	11	9	mTESE
[88]	1	1	100	1	1	1	cTESE
[185]	184	80	43	90	31	24	mTESE
[31]	76	31	40	44	18	21	cTESE
Totals	1809	777		669	313	315	

N: number of cases; P: Klinefelter syndrome patients; SSR: successful spermatozoa recovery; SRR: spermatozoa retrieval rate; ETC: embryo transfer cycle; CP: clinical pregnancy; NB: newborn; TESE: testicular sperm extraction; cTESE (37 cases): conventional TESE; mTESE (18 cases): microsurgical testicular sperm extraction; mixed (5 cases): cTESE+mTESE; FNA (2 cases): fine needle aspiration; TESA (1 case): testicular sperm aspiration; DTB: diagnostic testicular biopsy; y: age in years; total SRR: 777/1809 (43); cTESE: 228/516 (44%), range (16–100); mTESE: 427/991 (43%), range (17–100); CP rate: 313/669 (47%); NB rate: 315/669 (47%).

## Data Availability

This review was based on published data. Readers can request the PDF of the articles and the Excel document corresponding to Table 1.

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
