# Peer review of "The Klinefelter Syndrome and Testicular Sperm Retrieval Outcomes"

_genes, 2023, doi:10.3390/genes14030647_

Round 1

Reviewer 1 Report

·      This review article discussed the effects of Klinefelter syndrome on sperm retrieval from the testicles for subsequent use in assisted reproduction treatments. Also mentioned the predictive factors for successful spermatozoa retrieval and the risks of transmission of the genetic defect to children are also discussed.

·      The manuscript is clear, relevant to the field and presented in a well-structured manner. 

·      The cited references are relevant, but most are not recent publications (within the last five years). 

·      It includes some self-citations, but they are relevant to the field.

  • The statements and conclusions are drawn coherently and supported by the listed citations.

·      Overall, this exciting review adequately mentions and discusses the relevant literature. However, I would suggest several modifications, given that the paper has no Figures/Tables.

·      For the benefit of readers, the authors should include a Table systematically listing all the studies they cited, what they focused on and concluded, and how many patients were involved in each case. These should be separated by primary studies and meta-analyses, or just the latter if that is considered equally informative.

·      The plagiarism is relatively high and should be reduced enough to be accepted, especially expressions from DOI: 10.1111/j.2047-2927.2014.00231.x; DOI: 10.5935/15180557.20210081; DOI: 10.1210/jcem.83.1.4480)

·      Reproductive counselling and decision-making might be more clearly described in the text for the reader's benefit. The advantages and disadvantages of early fertility management need to be discussed. It is also noteworthy that the FDA has not authorized the use of Testosterone in patients less than 18 years old. Talking to these men as young adults about their fertility and the function that mTESE may play in preserving their fertility is crucial. The manuscript should also emphasise the strategies to optimize sperm retrieval rates and the optimal timing for sperm and testicular tissue cryopreservation. 

Author Response

Dear Reviewer 1:

We thank the Reviewer to have so detailed help us to improve the present manuscript.

We have included all your suggestions and answered to all your questions.

Reviewer 2 Report

Comments to the Authors

The Klinefelter syndrome and testicular sperm retrieval outcomes-Genes 2168769

The review manuscript gave a broad background on Klinefelter’s syndrome; the phenotypic presentations at histomorphological, molecular and physico- symptomological levels were highlighted. The manuscript also made mention to the importance of sperm retrieval as one of the techniques that can be used to treat infertility in patients with Klinefelter’s syndrome. Although more than 15 systematic and narrative reviews have been carried out on this subject matter, nevertheless, the detailed overview of the differences between mTESE and cTESE made this study different. Every point needed to make this manuscript a great review article is present. However, the arrangement, structure and delivery are very flawed. Before addressing an issue in a new paragraph, a brief overview should be given in the preceding paragraph. This will give the reader a sense of flow.

Herewith are my suggestions:

Abstract

From the abstract, ‘which’ should be added to the following sentence “Klinefelter syndrome (KS) is the most common chromosomal sexual anomaly, which has an”

The result described here is incidence not prevalence “estimated prevalence of 1:500/1000 per live born males, being due to the presence of an extra X

Infertility is not a characteristic; it is a disease condition caused by the syndrome. Kindly rephrase the sentence.

Infertility should rather be added to this list “KS has multisystemic consequences, endocrine (osteoporosis, obesity, diabetes), musculoskeletal, cardiovascular, autoimmune disorders, cancer and neurocognitive disabilities”

Remove the highlighted “comma” and “and” “Beyond, the hormonal and testicular histology characteristics, and the causal theories are discussed.”

The aim of the review is nicely highlighted in the abstract, however, there are enormous grammatical errors which must be corrected.

Main text:

The opening paragraph lacks coherence. Arrange the idea systematically and use fewer words.

Klinefelter syndrome (KS) is a frequent male genetic condition” this phrase passes no meaning... what kind of genetic condition is it?

This sentence is not complete and does not confer expected meaning “The hypergonadotropic hypogonadism derives from a structural and functional dysfunction of the testicular Leydig cells (LC) and Sertoli cells (SC), and of the hypothalamic-pituitary-gonad (HPG) axis”. Kindly re-phrase and restructure to convey the necessary meaning.

Merge the following sentences with previous sentence to convey meaning and reduce the use of unnecessary words. "A product" doesn't seem appropriate here. They are rather released from the HPG-A This leads to increased serum levels of the gonadotropins follicular stimulating hormone (FSH) and luteinizing hormone (LH) (a product of the hypothalamic-pituitary axis), and decreased of the androgen testosterone (T) levels (a product of the LC in response to LH) [17-19]”.

As the LC do not properly respond to LH, and SC do not respond properly to FSH and T, there is no negative feedback over the hypothalamic-pituitary axis” - This process does not correspond to the previous hypergonadotropic hypogonadism explained above. The negative feedback is rather inhibited because of the increase of serum FSH and LH. The increase in FSH and LH will inhibit the posterior hypothalamus to release GnRH and this will in turn inhibit the release of FSH and LH from the anterior pituitary. Hence, the above phenomenon described by the authors (As the LC do not properly respond to LH, and SC do not respond properly to FSH and T, there is no negative feedback over the hypothalamic-pituitary axis, LC become hyperplastic and the cells of seminiferous tubules responsible for spermatogenesis (SC and germ cells) degenerate)) can then follow.

A national evaluation in Denmark of 200 patients with”- the sentence should be restructured. You can link along these lines..... "For instance, findings from a national survey carried out on 200 patients with KS in Denmark, showed that ......"

“…growth and/or behavioral problems, and the remaining 45% were diagnosed in adulthood, mainly as part of infertility evaluation [21].” Section 2 can be inserted here and merged with this paragraph.

These patients evidence increased serum levels of FSH- evidence versus have

Signs and symptoms are variable, are not always present, and their appearance depend also on the

patient age”- connect the highlighted ideas properly.

Why are these letters in bold font? age, newborn characteristics, dysmorphic signs, congenital anomalies, hypotonia, genital anomalies, longer legs, speech disabilities, small and hard testes, etc.

The entire manuscript should be proofread for discrepancies in grammar, tense, idea structure, font-size and using bold letters inappropriately.

This syndrome presents genetic variability and endocrine consequences (osteoporosis, metabolic syndrome, obesity and type 2-diabetes)”- versus This syndrome presents genetic variability and endocrine consequences such as osteoporosis, metabolic syndrome, obesity and type 2-diabetes. Then start another sentence.

“…disturbances) [4, 5, 6, 20, 23, 46-50]. Regarding metabolic disorders”- These ideas are not connected nicely. Kindly connect each ideology coherently.

2. Prevalence and incidence”- There is no need to have a subheading for this. it can be merged with the paragraph that described the incidence rate in Denmark.

KS is the most common chromosomal sexual anomaly and the most common chromosomal anomaly in males’- Provide a reference for this sentence

“..be about 0.15% (150:100,000 males) of males in the general population”- Which specific population?

‘3. Hormonal and histological characteristics’ - Why the sudden jump to hormonal and histological characteristics? Kindly allow for flow of content. Let the reader be informed that you will be discussing the histological and hormonal status seen in KS and how these pathologies affect their reproductive potential.

Always define a term before the use of abbreviation

‘A dysfunction of LC, SC and of the HPG axis explains how the LC and SC fail to produce T, INSL3,

inhibin B and AMH in response to the increased levels of FSH and LH’- Too short to be a paragraph. It's an addendum to the previous sentence.

Etiology and phenotype variability should rather come before hormone and histological characteristics and effects

6. Spermatogenesis’ - There is a sudden jump. Authors needs to work on connecting ideas.

Why describing spermatogenesis at this point? Shouldn't spermatogenesis come before the histological characteristics of the seminiferous tubules?

9. Techniques of testicular sperm retrieval’- The authors elaborated greatly on the two types of TESE. However, only these two methods are mentioned. How about other methods of sperm retrieval? such as MESA, PESA, etc?

11. Conclusion’ - The entire conclusion section should be corrected for grammatical errors. No meaning is derived from the conclusion. I understand the layers of information that were described, however, the grammatical tenses used rendered the conclusion section hard to read. 

There is no need to include references in your conclusion. It is the conclusion of your study. It should describe the authors opinion about the topic and how the condition can be improved.

Recommendation for future studies should also be highlighted.

This is not an acknowledgment. If there is no one or institution to be acknowledged, then the sentences in the acknowledgement section should be deleted or rather say "Not available"

The table (Table 1) and the figure (Figure 1) are not included in the main manuscript. This should be inserted appropriately. 

Author Response

Dear Reviewer 2:

We thank the Reviewer to have so detailed help us to improve the present manuscript.

We have included all your suggestions and answered to all your questions.

Round 2

Reviewer 1 Report

My personal assessment is that this manuscript addresses a timely and interesting topic for physicians and surgeons. Your manuscript can be accepted as it stands.

Author Response

Dear Reviewer,

Thank you very much for your kindness to us.

Reviewer 2 Report

The Klinefelter syndrome and testicular sperm retrieval outcomes-Genes 2168769-R2

The entire manuscript has been improved significantly, both scientifically and grammatically. The readability of the abstract section has also improved drastically. Authors went straight to the point and the purpose for conducting the study was nicely portrayed. Howbeit, minor adjustment needs to be performed.

Minor comments

Although the current readability of the manuscript has improved significantly, authors should divide ideas instead of constructing longer sentences. Hence, the entire manuscript should be thoroughly proofread.

Line 25- will be discussed versus was discussed.

Line 38- add a comma, after chromosomes.

Kindly rephrase lines 46-49. Every information is there, but it is difficult to read.

Line 59 and 60- Hypergonadotropic hypogonadism is derived from…

The headings numbering should be verified. Shouldn’t “Predictive factors of testicular sperm retrieval” be 6.1 instead of 7, and so on?

Authors contribution section should be revised. The appropriate justification as to how each author contributed to the study should be provided. Providing micrographs or images does not justify becoming an author. The Journal’s guidelines should be followed for rationale to be included as an author.

Author Response

Dear reviewer,

As requested, all your suggestions were introduced.
